# An Integrated Nitrogen Management Strategy Promotes Open-Field Pepper Yield, Crop Nitrogen Uptake, and Nitrogen Use Efficiency in Southwest China

Xiao Ma [1,2], Fen Zhang [1,2], Fabo Liu [1,2], Guangzheng Guo [1,2], Taihong Cheng [1,2], Junjie Wang [1,2], Yuanpeng Shen [1,2], Tao Liang [1,2,3], Xinping Chen [1,2] and Xiaozhong Wang [1,2,*]

1   College of Resources and Environment, Southwest University, Chongqing 400716, China; mx1328@email.swu.edu.cn (X.M.); cyx970726@email.swu.edu.cn (F.Z.); ai19970307@email.swu.edu.cn (F.L.); zhanjswu@email.swu.edu.cn (G.G.); xiaohongmao@email.swu.edu.cn (T.C.); mahaoyue1@email.swu.edu.cn (J.W.); s13215709913@email.swu.edu.cn (Y.S.); zsh112233@email.swu.edu.cn (T.L.); chenxp2017@swu.edu.cn (X.C.)
2   Interdisciplinary Research Center for Agriculture Green Development in Yangtze River Basin, Southwest University, Chongqing 400716, China
3   Chongqing Academy of Agriculture Sciences, Chongqing 400000, China
*   Correspondence: wxz20181707@swu.edu.cn; Tel.: +86-1762-341-6568

**Abstract:** Excessive nitrogen (N) fertilizer application is a serious issue in intensive vegetable production and can negatively affect vegetable productivity and N use efficiency (NUE). The optimization of the N fertilizer rate and application of enhanced efficiency N fertilizers (EENFs), including nitrification inhibitors (Nis) and controlled-release fertilizer (CRF), are widely recognized as feasible N management strategies to resolve the problem of unreasonable N fertilizer input. Therefore, we conducted a 2-year field experiment (2019–2020) in an open-field vegetable system (pepper, *Capsicum annuum* L.) in southwest China to investigate the effects of an optimized N application rate and EENFs on vegetable yield, NUE, and crop N uptake. The following N management treatments were established: control without N fertilizer input (CK); optimized N fertilizer rate as urea (OPT); farmers' fertilizer practice (FP); application of a nitrification inhibitor (NI) within the optimized N fertilizer rate; and application of controlled-release fertilizer (CRF) within the optimized N fertilizer rate. The results showed that the OPT treatment based on root zone N management achieved a 37.5% reduction in the N application rate without compromising vegetable yield and increased the recovery efficiency of N (REN) by 31.5% compared to the FP treatment. Furthermore, the combined application of the NI or CRF treatments with the OPT treatment resulted in greater vegetable yields, fruit N uptake, and REN (9.54%, 26.8%, and 27.6%, respectively, for NI; 10.5%, 28.7%, and 28.8%, respectively, for CRF) than the OPT treatment alone. The absorption ratio of fruit N uptake to total crop N uptake was also increased. Our results clearly showed that the combined application of EENFs with the OPT treatment could achieve the win–win benefits of a yield increase and improved REN in Chinese vegetable production.

**Keywords:** open-field vegetable; yield; nitrogen-use efficiency; optimized N rate nitrification inhibitor; controlled-release fertilizer

## 1. Introduction

Vegetables provide food and nutritional security for the global population [1]. In China, the vegetable-planting area has reached approximately 24 million ha, accounting for 14.8% of the total farmland, and total vegetable production accounts for more than 50% of global vegetable production [2]. Intensive vegetable production is distinguished by high fertilizer inputs, particularly high nitrogen (N) application rates [3]. In a previous study, we found that vegetable production in China consumes 25% of the country's N fertilizer [4].

The N fertilizer rate per crop season is 364 kg N ha$^{-1}$, which is double that of grain crops and considerably exceeds the requirements of most vegetables [4,5]. This excessive N fertilizer rate will not only not increase the vegetable yield but will also result in a lower N use efficiency (NUE) and profits, creating a serious environmental burden [6–8]. The Chinese Ministry of Agriculture published the "Zero Increase Action Plan" for national fertilizer use in 2015, with the goal of improving fertilizer efficiency and maintaining high agricultural yields without additional fertilizer increases by 2020 [9]. There is an urgent need to establish a more optimized N management strategy to achieve sustainable vegetable production in China.

The "4R" management principle (right rate, right time, right place, and right source) of N fertilizer has been proposed to optimize N fertilizer management in vegetable production [10–13]. Optimizing the N application rate is the most fundamental step in achieving zero N fertilizer growth. Root zone N management is widely practiced in vegetable production systems to optimize N fertilizer use [14]. This management approach modifies N fertilizer application rates to achieve the synchronization of crop N requirements and N supply by considering the target N requirement and soil mineral N content in each crop growth period [15]. Previous studies indicated root zone N management practices can reduce N fertilizer inputs by 20–73% without compromising yield, compared to the conventional N management practices of local farmers in cucumber [16], tomato [17], and bitter gourd [18] systems. However, excessive N fertilizer rates can also cause serious reactive N losses due to the less-developed root system, weak nutrient uptake capacity, high fertilizer demand, and short growing period. The reactive N loss in vegetable systems increases linearly with an increase in N fertilizer dosage, which mainly results from the high soil N residue and high N fertilizer input [19,20]. This differs from the exponential relationship observed in grain crops [21]. Although the optimized N fertilizer dosage based on the root zone N management strategy can significantly reduce some of the reactive N loss, the recovery efficiency of nitrogen (REN) remains rather low [22], particularly in hot and rainy climatic conditions. New types of enhanced efficiency N fertilizers (EENFs), such as nitrification inhibitors (NIs) and controlled-release fertilizer (CRF), may be a good alternative to conventional urea to address this problem. A NI inhibits the nitrification reaction by affecting the activity of ammonia monooxygenase (AMO) and delays the conversion of ammonium N to nitrate N during nitrification [23], ensures the appropriate root zone nutrient concentration during the whole growing period of vegetables, and thus improves the vegetable yield and N uptake. Similarly, CRF synchronizes crop demand by effectively controlling the nutrient release rate [24]; thus increasing vegetable yield and nutrient use efficiency. NI and CRF can promote vegetable growth, increased N uptake and N use efficiency (NUE) and reduce reactive N loss compared to conventional urea application [25–28]. Furthermore, some recent meta-analyses have shown that the application of EENFs (i.e., NI and CRF) significantly improves vegetable yield and NUE [29–31]. The effects of NI and CRF on the NUE of crop yield varies among regions and crop types depending on the regional climate, soil, and field management practices [31,32]. Previous studies have focused on the effects of separate applications of NI and CRF, and optimizing the N dosage on crop yield, N uptake, and NUE, and there is a need to understand the combined effects of these management practices.

Southwestern China, which is a subtropical region characterized by high temperatures and high precipitation, is a major vegetable production region in China, accounting for 16% of national vegetable production. Pepper (*Capsicum annuum* L.) is one of the most representative vegetables in southwest China, where the overuse of N fertilizers has been frequently reported [33]. Therefore, a consecutive 2-year field trial was conducted to investigate the effects of EENFs with the optimal N rate on a comprehensive evaluation of yield, plant N uptake and NUE in southwest China. The results of this study will assist in the development of N management strategies to achieve sustainable vegetable production.

## 2. Materials and Methods

### 2.1. Experimental Site Information

Field experiments were conducted from 2019 to 2020 at the Hechuan Base of the Southwest University Experimental Farm (30°0′ N, 106°7′ E) in Chongqing Province, China. The cropping system in this experiment was a Chinese cabbage–pepper rotation system. Peppers are usually transplanted at the beginning of May and harvested at the end of August. The experimental site was located in a subtropical region characterized by high temperatures, and high but uneven precipitation. Based on measurements at a weather station close to the experimental site, the mean air temperature over the pepper cropping season was 25.0 °C in 2019 and 26.1 °C in 2020, and total precipitation over the pepper cropping season was 352 mm in 2019 and 289 mm in 2020 during the pepper cropping season (Figure 1).

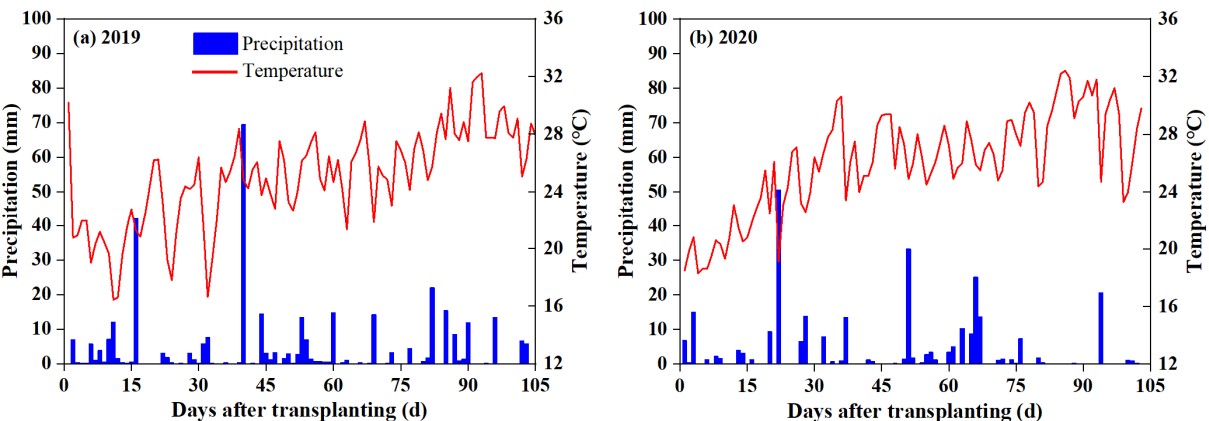

**Figure 1.** Daily mean temperature and precipitation during the pepper growing season in 2019 (**a**) and 2020 (**b**).

The cultivated soil in this study was classified as purplish soil. The main properties of the soil in the top 20 cm layer prior to the start of the experiment were as follows: pH, 5.65 (1:2.5, soil/water); total N, 0.50 g kg$^{-1}$; total P, 0.85 g kg$^{-1}$; total K, 24.3 g kg$^{-1}$; organic matter content, 9.19 g kg$^{-1}$; soil NO$_3^-$-N (extracted by CaCl$_2$), 4.89 mg kg$^{-1}$; soil NH$_4^+$-N (extracted by CaCl$_2$), 2.06 mg kg$^{-1}$; available P, 19.5 mg kg$^{-1}$; and exchangeable K, 99.9 mg kg$^{-1}$.

### 2.2. Experimental Treatments and Crop Management

A completely randomized block design, including five N treatments and four replicates, was established in 32 plots of 46.5 m$^2$ (5.6 m × 8.3 m). The five different N treatments were (1) control without N fertilizer input (CK); (2) optimized N fertilizer rate based on root zone N management (OPT, a conventional urea-N rate of 250 kg N ha$^{-1}$); (3) farmers' fertilizer practice (FP, a traditional urea-N rate of 400 kg N ha$^{-1}$); (4) application of an N stabilizer fertilizer containing 3,4-dimethylpyrazole phosphate as a nitrification inhibitor (NI, 250 kg N ha$^{-1}$ with ENTEC, produced by BASF, Ludwigshafen, Germany); (5) application of a controlled-release fertilizer (CRF, 250 kg N ha$^{-1}$ with polyurethane-coated urea, produced by the Maoshi Eco-Fertilizer company, Linyi, Shandong Province, China). The amount of phosphorus and potassium fertilizer for each treatment and fertilizer application rates for different periods are provided in Table 1. All fertilizers were applied by spot application around 10–15 cm below the soil surface near the plants.

**Table 1.** Fertilizer application rates at different growth stages under different treatments (kg ha$^{-1}$).

| Treatment | Fertilizer Application Rates (N-P$_2$O$_5$-K$_2$O) | | | | |
|---|---|---|---|---|---|
| | Seedling Period | Blooming and Fruit-Setting Period | Mid-Fruiting Period | Full-Fruiting Period | Total |
| CK | 0–70–60 | 0–70–80 | 0–0–80 | 0–0–80 | 0–140–300 |
| OPT | 100–70–60 | 50–70–80 | 50–0–80 | 50–0–80 | 250–140–300 |
| FP | 280–145–115 | 120–145–115 | 0–0–0 | 0–0–0 | 400–290–230 |
| NI | 100–70–60 | 50–70–80 | 50–0–80 | 50–0–80 | 250–140–300 |
| CRF | 250–70–60 | 0–70–80 | 0–0–80 | 0–0–80 | 250–140–300 |

In the 2019 experiment, peppers were transplanted on 27 April, and the basal and three topdressing fertilizer applications were provided during the seedling period, blooming and fruit-setting period, mid-fruiting period, and full-fruiting period on days 13, 48, 66, and 87 after transplanting, respectively. In the 2020 experiment, peppers were transplanted on 12 May, and the basal and three topdressing fertilizers were applied in the seedling period, blooming and fruit-setting period, mid-fruiting period, and full-fruiting period on days 12, 41, 64, and 87 after transplanting, respectively. The fertilizer application rates in the different periods for each treatment are shown in Table 1. Peppers were harvested three times in 2019 on 6 July, 25 July, and 9 August and twice in 2020 on 20 July and 22 August. In both years, peppers were cultivated with a row spacing of 60 cm and a within-row plant spacing of 40 cm. All other field management was conducted in accordance with local practices.

*2.3. Sample Collection and Analysis*

In both years, the peppers were picked from 24 plants within the 2 middle rows of each plot and weighed at each harvest. The total yield was calculated as the cumulative weight of peppers from all harvest days. Three physically similar plants with almost identical in-field growth were taken at the seedling stage, blooming and fruit-setting stage, mid-fruiting stage, full-fruiting stage and ripening stage, respectively. The stem, leaf, and fruit were collected separately and then dried at 75 °C until they were completely dry, before the dry mass of each part was weighed. The dry samples of different organs were subsequently ground into a powder for determination of the N concentration (Kjeldahl procedure). The N accumulation was calculated using the formula TDM × NC, where TDM represents the total dry matter of the stem, leaf, and fruit, and NC represents the N concentration in the stem, leaf, and fruit. The N use efficiencies, such as the recovery efficiency of N (REN), the agronomic efficiency of N (AEN), and the partial factor productivity of N (PFPN) were calculated using the following formulas:

(1)   REN (%) = (total plant N accumulation in N application treatment − total plant N accumulation in CK treatment)/N rate × 100.

(2)   AEN (kg kg$^{-1}$) = (fruit yield in N application treatment–fruit yield in CK treatment)/N rate.

(3)   PFPN (kg kg$^{-1}$) = fruit yield in N application treatment/N rate.

*2.4. Statistical Analysis*

Analysis of variance (one-way ANOVA) was applied to determine the significance among the treatments. Multiple comparisons of Tukey tests were conducted to evaluate the variation from treatments and years. Treatment means were compared using the least significant difference (LSD) at the 0.05 probability level. All statistical analyses were performed using SPSS Version 20.0 (SPSS Inc., Chicago, IL, USA), and graphs were generated using Origin Version 2019b (OriginLab, Hampton, MA, USA).

## 3. Results

### 3.1. Yields

Applying N fertilizer significantly increased the pepper yield compared to the control treatment (Figure 2). The average total fresh fruit yields over the 2 years from the OPT, FP, NI, and CRF treatments were $36.4 \pm 0.56$, $37.4 \pm 1.81$, $39.9 \pm 0.60$, and $40.3 \pm 0.76$ t ha$^{-1}$, respectively. There were no significant differences in the pepper yield between the FP treatment and other optimization fertilizer treatments. Compared to the OPT treatment, the total fruit yields in the NI and CRF treatments were increased by 9.54% and 10.5%, respectively, whereas there were no significant differences in yield between the NI and CRF treatments. In addition, the yields of pepper in the 2019 and 2020 seasons significantly differed, with total yields in the 2019 season being 11.7% lower than in the 2020 season. This difference was attributed to infection by soft rot due to continuous high temperature combined with continuous precipitation at the end of the growing season in 2019 (Figure 1).

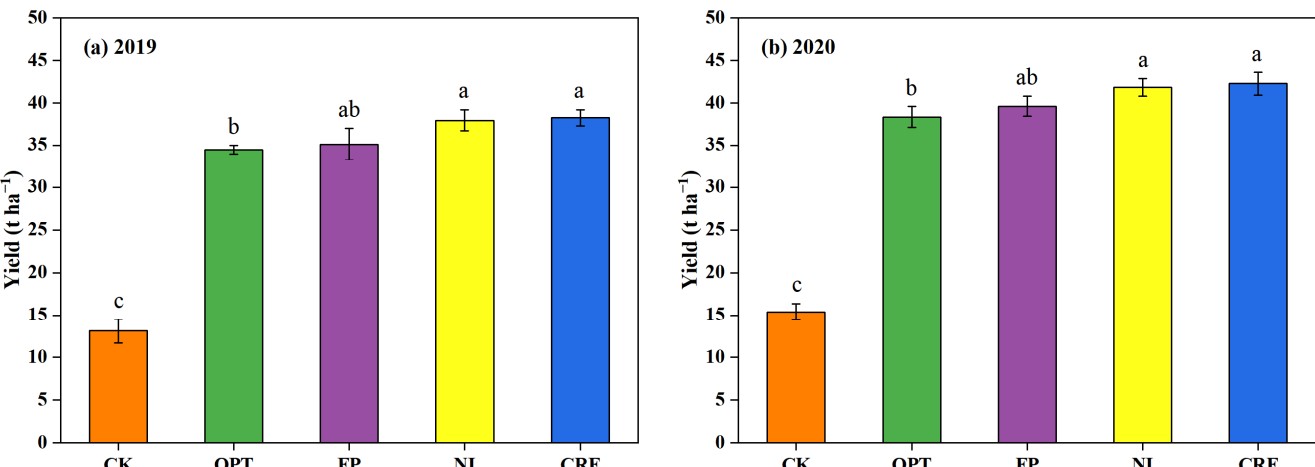

**Figure 2.** Fruit yield of pepper under different N management strategies in 2019 (**a**) and 2020 (**b**). Different lowercase letters indicate statistical significance at the *p* = 0.05 level. The vertical bars represent the standard error (*n* = 4).

### 3.2. Biomass Accumulation and Partitioning

The dry weight (DW) accumulation was measured at different growing stages of the pepper crop as shown in Figure 3. The dynamic process of DW accumulation followed the "slow–fast–slow" rule. The total DW of pepper was lowest in the CK treatment in both years. The results of the 2-year field trials showed that the dry matter accumulation of pepper followed the trend of FP > CRF > NI > OPT in both years (Figure 3). Compared to the OPT treatment, the total dry weight of the FP, NI, and CRF treatments in the harvest period significantly increased by 10.9%, 5.93%, and 6.10%, respectively. In the harvesting period during 2019–2020, the mean DW of the vegetative organs (stem + leaf) in the FP treatment was significantly higher than in the OPT, NI, and CRF treatments (by 28.1%, 31.7%, and 34.0%, respectively) (Figure 4a). There were no significant differences in the mean DW of the vegetative organs (stem + leaf) among the three optimized fertilizer strategies. In the harvesting period, the mean DW of the fruits in the NI and CRF treatments was higher than in the FP treatment by 7.58% and 8.65%, respectively, and higher than in the OPT treatment by 10.2% and 11.3%, respectively (Figure 4b). There was no significant difference in the mean DW of the fruits between the NI and CRF treatments (Figure 4b). Compared to the FP and OPT treatments, the EENFs (NI and CRF) increased the average distribution ratio of fruits by 13.2% and 4.46% and reduced the average distribution ratio of vegetative organs by 27.0% and 9.91%, respectively (Figure 4b). Similarly, there were no significant differences in the distribution ratio of fruits and vegetative organs between the NI and CRF treatments.

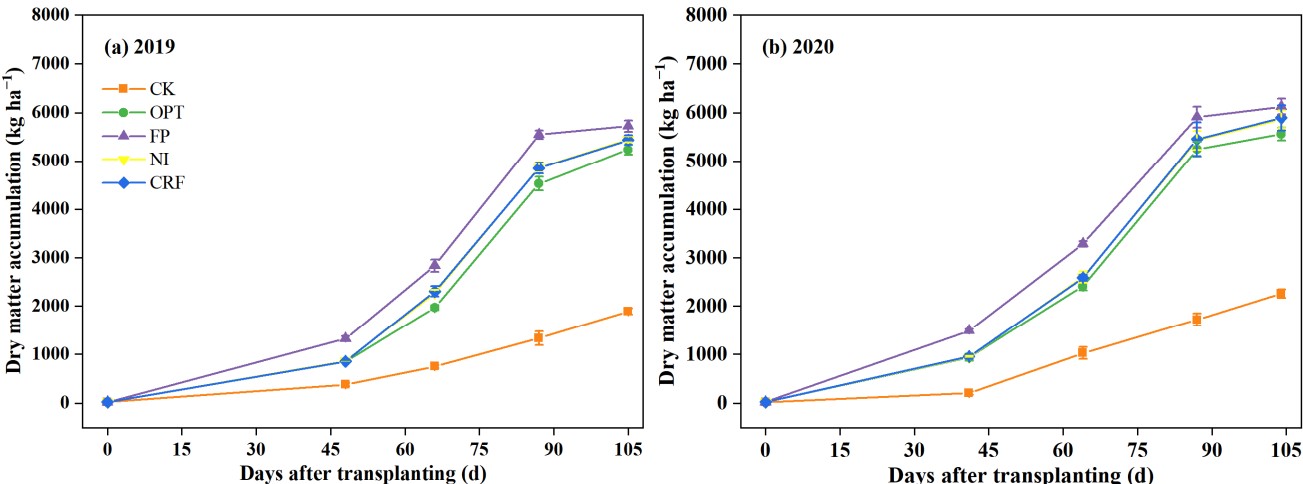

**Figure 3.** The dynamics of dry weight accumulation under different N management strategies in 2019 (**a**) and 2020 (**b**). The vertical bars represent the standard error (*n* = 4).

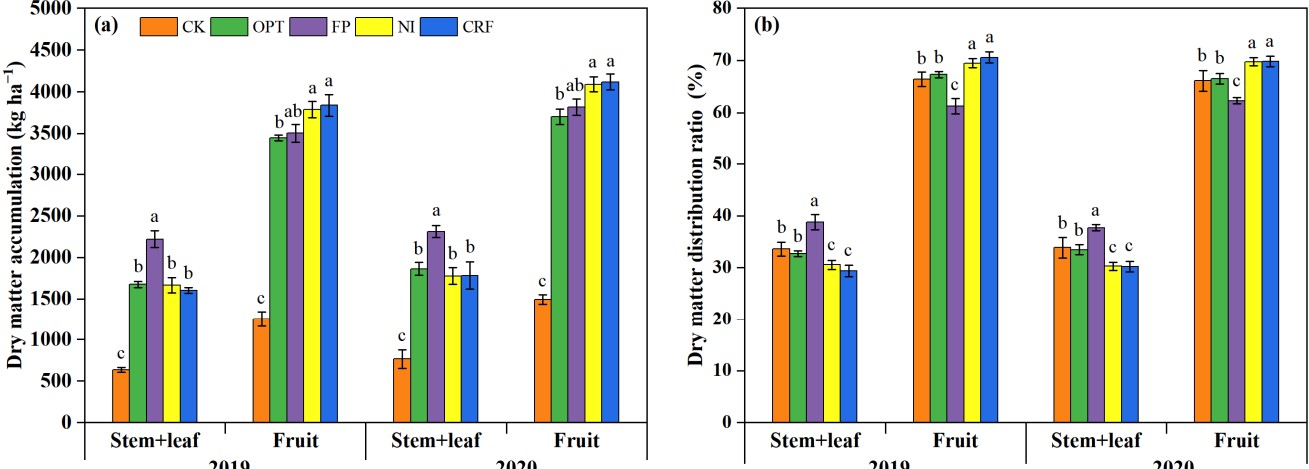

**Figure 4.** Amount (**a**) and proportion (**b**) of dry matter allocated in vegetative organs (stem + leaf) and fruit in the harvesting periods in 2019 and 2020. Different lowercase letters indicate statistical significance at the *p* = 0.05 level. The vertical bars represent the standard error (*n* = 4).

### 3.3. Nitrogen Accumulation and Partitioning

The dynamic process of N accumulation was similar to that of dry matter and followed the "slow–fast–slow" rule (Figure 5). Compared to the OPT treatment, the average total N accumulation in the harvest period from the FP, NI, and CRF treatments significantly increased by 15.4%, 19.6%, and 20.5%, respectively, with no statistical differences observed among the FP, NI, and CRF treatments. In the harvesting periods during 2019–2020, the average total N accumulation in vegetative organs (stem + leaf) in the FP treatment was significantly higher than in the OPT, NI, and CRF treatments by 29.5%, 22.6%, and 23.9%, respectively (Figure 6a). There were no significant differences in the average N accumulation of the vegetative organs (stem + leaf) among the three optimized fertilizer strategies. In the harvesting period, the average total N accumulation in fruit in the NI and CRF treatments was greater than in the FP treatment by 17.5% and 19.3%, respectively, and greater than in the OPT treatment by 26.8% and 28.7%, respectively (Figure 6b). There were no significant differences in the average N of the vegetative organs (stem + leaf) in the NI and CRF treatments, both of which were significantly higher than in the OPT and FP treatments. Compared to the FP and OPT treatments, the EENF (NI and CRF) treatments increased the distribution ratio of fruits by 28.2% and 14.1% and reduced the distribution ratio of vegetative organs by 13.9% and 6.44%, respectively (Figure 6b). Similarly to DW,

there were no significant differences in the distribution ratio of fruits and vegetative organs between the NI and CRF treatments.

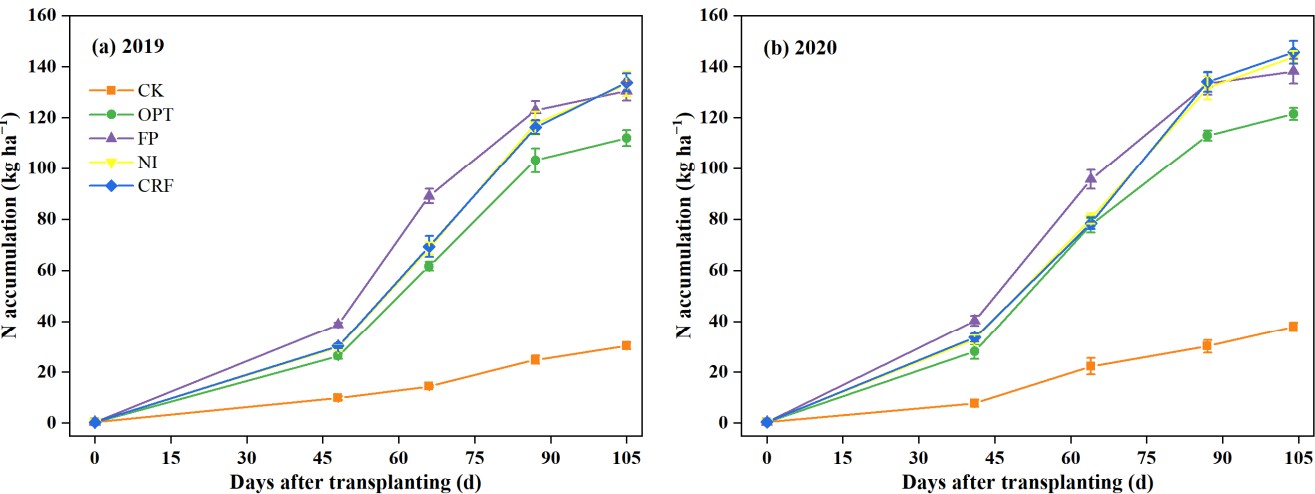

**Figure 5.** Dynamics of pepper N accumulation under different N management strategies at different growth periods of pepper in 2019 (**a**) and 2020 (**b**). The vertical bars represent the standard error (*n* = 4).

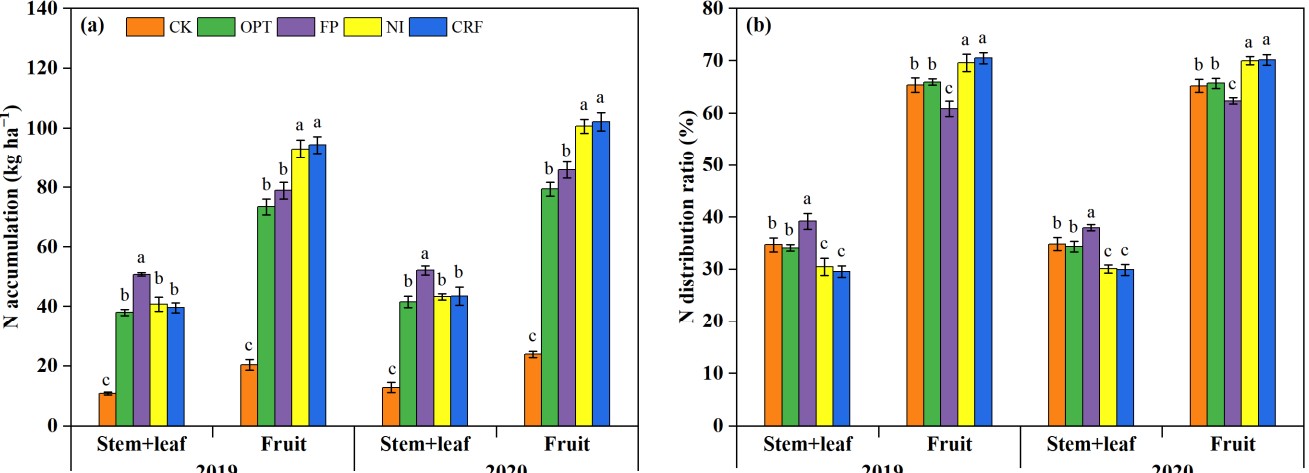

**Figure 6.** Amount (**a**) and distribution ratio (**b**) of N allocated to the vegetative organs (stem + leaf) and fruit in the harvest period in 2019 and 2020. Different lowercase letters indicate the statistical significance at the *p* = 0.05 level. The vertical bars represent the standard error (*n* = 4).

### 3.4. Nitrogen Use Efficiencies

Compared to the FP treatment, the OPT, NI, and CRF treatments increased the PFPN by 55.9%, 70.8%, and 72.4% over the 2 years; increased the AEN by 53.4%, 77.5%, and 80.0%; and increased the REN by 31.4%, 67.7%, and 69.4%, respectively (Table 2). Compared to the OPT treatment, the NI and CRF treatments significantly increased the REN by 27.6% and 28.8%, respectively (Table 2). There were no significant differences in PFPN, AEN, and REN between the NI and CRF treatments.

**Table 2.** Effects of different treatments on the N use efficiencies of pepper in 2019 and 2020.

| Year | Treatment | PFPN (kg kg$^{-1}$) | AEN (kg kg$^{-1}$) | REN (%) |
|---|---|---|---|---|
| | CK | — — | — — | — — |
| | OPT | 138 a | 85.6 a | 32.1 b |
| 2019 | FP | 87.8 b | 55.1 b | 24.7 c |
| | NI | 152 a | 99.4 a | 41.1 a |
| | CRF | 153 a | 101 a | 41.2 a |
| | CK | — — | — — | — — |
| | OPT | 153 a | 91.8 a | 33.7 b |
| 2020 | FP | 99.0 b | 60.6 b | 25.4 c |
| | NI | 167 a | 106 a | 42.9 a |
| | CRF | 169 a | 108 a | 43.6 a |
| | Analysis of variance | | | |
| Year | | * | * | NS |
| Treatment | | * | * | * |
| Year × Treatment | | NS | NS | NS |

NS, not significant; * significant at the *p* = 0.05 level. Different lowercase letters indicate the statistical significance at the *p* = 0.05 level.

## 4. Discussion

In recent years, many studies have shown that it is possible to reduce N fertilizer inputs to achieve sustainable vegetable development [34–37]; while increasing vegetable yields remains a great challenge. In our study, the OPT treatment (urea-N rate of 250 kg N ha$^{-1}$ based on root zone N management) did not cause yield losses based on a 37.5% reduction in N fertilizer use compared to the FP treatment, similar to previous studies [36,37]. This was because farmers in the study area applied more than 50% of the total N fertilizer in the basal fertilizer, which resulted in excessively heavy vegetative growth in the early stages of pepper growth and insufficient N supply in the later stages of pepper growth, limiting the translocation and distribution of biomass to the fruit. The OPT treatment reduced the amount of basal fertilizer N application and increased N topdressing twice in the middle and full fruit stages to ensure an appropriate N supply throughout the whole pepper growing period to achieve a quantitative match between N demand and N supply. Therefore, the OPT treatment reduced the unnecessary vegetative growth compared to the FP treatment to some extent and optimized the partitioning pattern of dry matter and N. This resulted in more dry matter and N being accumulated in the fruit, which further increased the pepper yield and was consistent with previous studies in bitter gourd [18] and maize [38]. In addition, the OPT treatment significantly improved the REN, AEN, and PFPN, mainly by maintaining the crop yield and crop N uptake at a lower N fertilizer rate. This is consistent with the results of previous studies on grain crops [39] and other vegetable crops [37]. Therefore, optimizing the N fertilizer rate based on a region-specific and crop-specific root zone N management strategy is necessary.

The application of EENFs (NI and CRF) is another important measure that can be used to achieve sustainable vegetable production. We found that the application of NI and CRF significantly increased yields by 9.54% and 10.5%, and REN by 27.6% and 28.8%, respectively, compared to the conventional urea treatment at the same N application rate, which was consistent with the results of previous studies [11,40,41]. In addition, a meta-analysis of previous studies showed that the application of CRF significantly increased vegetable yields by 7% and NUE by 11% [31], and NI significantly increased vegetable yields by 7% [29]. The increased vegetable yield and improved NUE of NI and CRF were due to the nitrification inhibitor effectively inhibiting ammonium N nitrification compared to urea, reducing N loss, and maintaining a high ammonium N concentration in the root zone for a long time. The CRF treatment could effectively control the rate of N release and maintain a higher N supply, both of which can better synchronize N supply with pepper demand at temporal and spatial scales [42], promote middle and late pepper growth, and increase the proportion of N distributed from stems and leaves to fruits after flowering

and fruiting. Thus, compared to the conventional urea application, NI and CRF further improved the vegetable N uptake, yield, and N fertilizer use efficiency.

Producing more vegetables will ensure the supply of vegetable food products and meet human health requirements during periods of economic and population growth. Excessive N fertilizer rates are commonly encountered in the current intensive vegetable production systems and will affect vegetable crop yield. Many previous studies have indicated that an optimized N fertilizer rate based on crop N demand and soil N residue will slightly increase or maintain vegetable yield [6,35]. Our study demonstrated that combining the application of EENFs (NI and CRF) with the optimal N rate can achieve a "win–win" of increased vegetable yields, crop N uptake, and REN, and our study provided good support for managing vegetable crops in other areas. The sustainable intensification of agriculture systems is needed to support the growing population in China over the next few decades [43], and therefore crop yield must substantially increase. Therefore, the IKPS strategy (i.e., integrated knowledge of the best nutrient and crop management strategies combined with use of the most effective products) has been proposed [43]. The IKPS strategy has been applied in 54 field trials across the country, including 13 common vegetable varieties. The results have shown that IKPS-based management reduces the N fertilizer rate by 38% and increases the vegetable yield and N uptake by an average of 17.3% and 12.8%, respectively, compared to local farmers' practices. Because the IKPS conceptual framework and methodological principles need to be adapted to local soil and climatic conditions for specific measures in different regions, achieving sustainable vegetable production in the future needs to be verified in southwest China.

## 5. Conclusions

It is vital to achieve sustainable vegetable production in China and elsewhere in the world. A 2-year field trial indicated that an optimized N fertilizer rate based on root N management could maintain pepper yield and increase NUE by 5%, with a 37.5% reduction in N fertilizer application compared to the conventional practices of local farmers. Compared to a conventional urea treatment at the same optimized N application rate, a nitrification inhibitor and controlled release fertilizer improved yield, crop N uptake, and NUE. This study established a sustainable and promising N fertilizer management strategy for an open-field pepper production system in southwestern China.

**Author Contributions:** Conducted research work, collected and analyzed the data, and prepared the manuscript, X.M., F.Z., F.L., G.G., T.C., J.W. and Y.S.; designed, supervised this study, revised and approved this manuscript for publication, T.L., X.C. and X.W. All authors have read and agreed to the published version of the manuscript.

**Funding:** This work was support by the National Natural Science Foundation of China (U20A2047); Changjiang Scholarship, Ministry of Education, China; and State Cultivation Base of Eco-agriculture for Southwest Mountainous Land, Southwest University.

**Conflicts of Interest:** The authors declare no conflict of interest.

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
