# Peer review of "An Integrated Nitrogen Management Strategy Promotes Open-Field Pepper Yield, Crop Nitrogen Uptake, and Nitrogen Use Efficiency in Southwest China"

_agriculture, doi:10.3390/agriculture12040524_

Round 1
Reviewer 1 Report
The manuscript by Ma et al. is generally well written, fairly easy to follow and has conclusions justified by the results. Of particular interest was the inclusion of both time-release and nitrification-inhibitor fertilizer formulations for comparison to fertilization done by traditional farming practices (at least as they apply to peppers in the region of China studied). My specific comments are:
- Table 1 appears to be missing, so I lack information on fertilizer application rates.
- I think the reference to Figure 1 in line 172 actually means Figure 2.
- Abstract, Line 16, might be better to change "indiscriminate" to "excessive".
- Figure 2. Is the yield per hectare extrapolated from the several plants analyzed in these experiments (n = 4)?
- Figures 3 and 5: the yellow lines (used for NI) are nearly invisible, at least on my monitor. They seem to follow closely the curves for CRE, and I based my assessment of the data on this.
- Is the "slow-fast-slow" rule "logical" as stated in line 187, or just the "expected" trend. I think it's the latter.
- Last lines of the Discussion, lines 301-302, are weak, stating the "possibility" of something (sustainable vegetable production) "needs to be verified". Perhaps something based on the last sentence of the Conclusions would be a better choice because it is based on what was found in the study.
Author Response
Reply to Reviewer #1
Thank you all for the valuable comments and suggestions concerning our manuscript entitled“An integrated nitrogen management strategy promotes open-field pepper yield, crop nitrogen uptake, and nitrogen use efficiency in southwest China”(ID: agriculture-1617942). We have made great efforts to revise the manuscript in response to these comments and suggestions. The detailed changes in the manuscript text were highlighted in red color, and the details of changes and revisions were listed below for your reference.
Sincerely,
Xiaozhong Wang
Question #1:Table 1 appears to be missing, so I lack information on fertilizer application rates.
Response: Yes, you are right. We have supplemented Table1 in the manuscript. Please see line 173.
Question #2:I think the reference to Figure 1 in line 172 actually means Figure 2.
Response: Thanks. We have revised it. Please see line 180.
Question #3:Abstract, Line 16, might be better to change "indiscriminate" to "excessive".
Response: Yes, you are right. We have substituted "indiscriminate" with "excessive" in line 16.
Question #4:Figure 2. Is the yield per hectare extrapolated from the several plants analyzed in these experiments (n=4)?
Response: Thank you for your comments. The n=4 in this sentence represents the number of replicates of our experimental treatment. The peppers were picked from 24 plants within the 2 middle rows of each plot and weighed at each harvest. The total yield was calculated as the cumulative weight of peppers from all harvest days (The more details are listed in line 148-150).
Question #5:Figures 3 and 5: the yellow lines (used for NI) are nearly invisible, at least on my monitor. They seem to follow closely the curves for CRE, and I based my assessment of the data on this.
Response: Many thanks. In the figure below we tried replacing the yellow line with a red one, but it is still not very visible. The yellow line (used for NI treatment) is almost invisible due to the high degree of overlap with the blue line ( used for CRF treatment).
Question #6:Is the "slow-fast-slow" rule "logical" as stated in line 187, or just the "expected" trend. I think it's the latter.
Response: Agree. To avoid confusion, we have deleted "logical".
Question #7:Last lines of the Discussion, lines 301-302, are weak, stating the "possibility" of something (sustainable vegetable production) "needs to be verified". Perhaps something based on the last sentence of the Conclusions would be a better choice because it is based on what was found in the study.
Response: Sorry for the inappropriate word. We have deleted the word and revised the sentence in the revised vision. Please see line 314-315.

Reviewer 2 Report
The authors reported different nitrogen management strategies in open-field pepper yield. The results showed that application of enhanced efficiency N fertilizers (EENFs) increase pepper yield and improve N use efficiency in pepper production in China.
The document has some weakness such as few details in the statistical analysis, mistakes such a table that is not present in the document. Here there are some comments and suggestion in order to improve this manuscript.
Introduction:
Line 49. Write N use efficiency before NUE
Line 92. The Latin name of pepper must be in italic font.
Methods:
Line 114. The Organic matter is just 0.9% it is very low. Do you think that could be affect your experiment in soil with highest OM. Maybe is better to try to find alternatives to increase OM in the soil that should help to don’t lose too much N in the crop system.
Line125 and 126. What is the name of the product, you should put this information if someone what to replicate your experiment.
Line 138. There is not a Table 1 in the document.
Line 139. How do you can compare yield between years if the authors did different amount of harvests in each year. And seems at 2020 that the authors did two harvest had more yield that in 2019. Later, in the results the authors explain that was due to blossom end rot. This problem in vegetables is due to deficient of Ca. If the authors did the experiment in the same location in both years why it was not see this problem in 2020. Did the authors do an complementary fertilization with Ca?
Line 148. This sentence is confuse how many stages in total was sampling for biomass accumulation. There are too many “and “ in the sentence and that makes it confuse.
Line 162. Did the authors run a ANOVA first to do DUNCAN test? They should give more details about the statistical analysis.
Results:
Line 189. It is not Figure 2, it is supposed to be Figure 3.
Line 196-204 and 219-230. The % differences amoung treatments that the authors mentioned are related of the average of both years?
Discussion:
Line 248. Should add OP (urea-N rate of 250 kg N ha-1 in root area) ,
Could the authors explain more if there are any ecological advantage or disadvantage to use EEFNs in the soil. Seems for economical beneficial the farmers could just apply the optimal practice (250 kg N ha-1 in root area) and don’t expend more money in EEFNs, I don’t have idea how much could a farmer spend in EEFNs but this value could justify the 10% of the yield increased?
Reference:
Al the references must be in the same format. For example references 9, 38 and 21 all the words for the title start with capital letters and the rest of the references not. Besides the Latin names of the plant species must be in italic font.
Author Response
Reply to Reviewer #2
Thank you all for the valuable comments and suggestions concerning our manuscript entitled“An integrated nitrogen management strategy promotes open-field pepper yield, crop nitrogen uptake, and nitrogen use efficiency in southwest China”(ID: agriculture-1617942). We have made great efforts to revise the manuscript in response to these comments and suggestions. The detailed changes in the manuscript text were highlighted in red color, and the details of changes and revisions were listed below for your reference.
Sincerely,
Xiaozhong Wang
Question #1:Line 49. Write N use efficiency before NUE
Response: Yes, you are right. Changes are made as suggested. Please see line 50.
Question #2:Line 92. The Latin name of pepper must be in italic font.
Response: Yes, you are right. Changes are made as suggested. Please see line 22 and line 96.
Question #3:Line 114. The Organic matter is just 0.9% it is very low. Do you think that could be affect your experiment in soil with highest OM. Maybe is better to try to find alternatives to increase OM in the soil that should help to don’t lose too much N in the crop system.
Response: Yes, you are right. We also think soil organic matter (SOM) has an impact on the results of our experiment. However, low SOM is a typical feature of open field vegetable production systems in Southwestern China. The reasons for this result are listed as follows: on one hand, high frequency of tillage operation and excessive chemical fertilizer rate for vegetable system in this region generally results in the breakup of soil macro-aggregates, soil structure damage, and an increase in soil aeration, which promotes the microbial decomposition of SOM. On other hand, the decomposition of SOM is accelerated by high air temperatures and soil moisture levels in southwest China. Our current study was carried out under the specific soil and climatic conditions in southwestern region. Meanwhile, we will continue to find alternatives to increase soil organic matter SOM, such as the application of organic fertilizers or biochar, on improving crop N uptake and reducing N losses in the future.
Question #4:Line125 and 126. What is the name of the product, you should put this information if someone what to replicate your experiment.
Response: Thank you. we have added the name of the product. Please see line 128-130.
Question #5:Line 138. There is not a Table 1 in the document.
Response: Many thanks. We have added table1 in the manuscript. Please see line 173.
Question #6:Line 139. How do you can compare yield between years if the authors did different amount of harvests in each year. And seems at 2020 that the authors did two harvest had more yield that in 2019. Later, in the results the authors explain that was due to blossom end rot. This problem in vegetables is due to deficient of Ca. If the authors did the experiment in the same location in both years why it was not see this problem in 2020. Did the authors do an complementary fertilization with Ca?
Response: Thanks for the reviewe’s suggestion. We have rechecked the growth of the peppers during whole field growing season and found the rotten fruit and the decrease of pepper yield are mainly attributed to soft rot, which is caused by continuous high temperature combined with continuous precipitation during the full-fruiting period, rather than blossom end rot. We have revised this in the manuscript. Please see Line 188-190.
Question #7:Line 148. This sentence is confuse how many stages in total was sampling for biomass accumulation. There are too many “and“ in the sentence and that makes it confuse.
Response: Yes, you are right. We have revised it. Please see line 148-150.
Question #8:Line 162. Did the authors run a ANOVA first to do DUNCAN test? They should give more details about the statistical analysis.
Response: Thanks for the reviewer’s suggestion. We have listed more details about the statistical analysis. Please see line 167-170.
Question #9:Line 189. It is not Figure 2, it is supposed to be Figure 3.
Response: Thank you. Changes are made as suggested.
Question #10:Line 196-204 and 219-230. The % differences amoung treatments that the authors mentioned are related of the average of both years?
Response: Thank you. The percentage differences between treatments mentioned in line 196-204 and 219-230 of the manuscript are the average of both years.
Question #11:Line 248. Should add OP (urea-N rate of 250 kg N ha-1 in root area)
Response: Thanks for your nice suggestion. Changes are made as suggested. Please see line 258-259 in the revised vision.
Question #12:Could the authors explain more if there are any ecological advantage or disadvantage to use EEFNs in the soil. Seems for economical beneficial the farmers could just apply the optimal practice (250 kg N ha-1 in root area) and don’t expend more money in EEFNs, I don’t have idea how much could a farmer spend in EEFNs but this value could justify the 10% of the yield increased?
Response: Thanks for your nice suggestion. Our results find out that relative to optimal practice, the application of EENFs could increase the pepper yield and nitrogen recovery efficiency by 9.54-10.5% and 27.6-28.8%, respectively, meanwhile although the cost of EENFS was higher than conventional fertilizer, the net benefits in EENFS treatment was higher by 3.39-9.77% than that in optimal practice through our economic analysis. In addtion, the application of EFFNs could significantly mitigate the reactive nitrogen loss. Our field experiments have also investigated the effect of EENFs on reactive N losses such as N2O and N leaching. By comparing the OPT, NI and CRF treatments, it was found that the NI and CRF treatments reduced N2O emissions by 77.2% and 37.7% respectively compared to the OPT treatment at the same N fertiliser application rate (Figure 1). In conclusion, comprehensive considering the yield, nitrogen use efficiency, economic benefits and ecological advantage, the the application of EFFNs is better for open-field vegetable system in Southwest region.
Figure 1. Cumulative seasonal N2O emission (kg N ha-1) during the pepper growth seasons from 2019 to 2020.
Question #13:All the references must be in the same format. For example references 9, 38 and 21 all the words for the title start with capital letters and the rest of the references not. Besides the Latin names of the plant species must be in italic font.
Response: Thanks for the reviewer’s suggestion. We have standardized the format of the references.
